# TS-RAG: Retrieval-Augmented Generation based Time Series Foundation Models are Stronger Zero-Shot Forecaster

## Abstract

Recently, Large Language Models (LLMs) and Foundation Models (FMs) have become prevalent for time series forecasting tasks. However, fine-tuning large language models (LLMs) for forecasting enables the adaptation to specific domains but may not generalize well across diverse, unseen datasets. Meanwhile, existing time series foundation models (TSFMs) lack inherent mechanisms for domain adaptation and suffer from limited interpretability, making them suboptimal for zero-shot forecasting. To this end, we present TS-RAG, a retrieval-augmented generation based time series forecasting framework that enhances the generalization capability and interpretability of TSFMs. Specifically, TS-RAG leverages pre-trained time series encoders to retrieve semantically relevant time series segments from a dedicated knowledge database, incorporating contextual patterns for the given time series query. Next, we develop a learnable Mixture-of-Experts (MoE)-based augmentation module, which dynamically fuses retrieved time series patterns with the TSFM's representation of the input query, improving forecasting accuracy without requiring task-specific fine-tuning. Thorough empirical studies on seven public benchmark datasets demonstrate that TS-RAG achieves state-of-the-art zero-shot forecasting performance, outperforming TSFMs by up to 6.51% across diverse domains and showcasing desired interpretability.

## 1. Introduction

Time series forecasting, which aims to predict future values of a sequence based on its past observations, plays a critical role in various real-world applications, *e.g.*, finance, healthcare, energy management, and climate science. The key idea is to capture the temporal dependency patterns in the form of trend, seasonality, autocorrelation, *etc*, to make accurate prediction and generalize across different datasets.

In the past, a substantial amount of effort has been made to tackle this problem. Traditional statistical methods such as AutoRegressive Integrated Moving Average (ARIMA) (Whittle, 1951) work well for stationary time series but struggle with complex dependencies and non-linear patterns. Machine learning approaches, such as Random Forest (Breiman, 2001) and XGBoost (Chen & Guestrin, 2016), can handle external covariates of features but fail to capture long-range dependencies. Deep learning techniques, including Long Short-Term Memory (LSTM) (Hochreiter & Schmidhuber, 1997), Gated Recurrent Units (GRUs) (Cho et al., 2014), Temporal Convolutional Networks (TCNs) (Lea et al., 2016), Graph Neural Networks (GNN) based models (Cao et al., 2020; Shang et al., 2021), and transformer based models (Vaswani et al., 2017; Zhou et al., 2021) are typically trained on specific domains and may not perform well to diverse unseen datasets. More recently, there is a prevalent interest in adapting Large Language Models (LLMs) (Brown et al., 2020; Touvron et al., 2023b; Achiam & et al., 2023) for time series tasks (Gruver et al., 2024; Jiang et al., 2024) and developing Foundation Models (FM) tailored for time series data (Rasul et al., 2023; Garza & Mergenthaler-Canseco, 2023; Liu et al., 2024b). Although both LLMs and FM have shown great promise in improving forecasting accuracy and handling complex temporal dynamics, they still face immense barriers when applied to zero-shot forecasting tasks, limiting their real-world applicability.

Specifically, fine-tuning LLMs for time series forecasting can enable the adaptation to specific datasets, however, those methods could struggle with generalization across diverse, unseen domains (Zhou et al., 2023; Jin et al., 2023; Pan et al., 2024). In addition, they typically involve heavy computational costs, even with limited samples. Time Series Foundation Models (TSFMs) (Das et al., 2023; Ekambaram et al., 2024; Ansari et al., 2024; Woo et al., 2024), while effective in learning general time series representations, lack inherent mechanisms for domain adaptation as they can-

[1]Anonymous Institution, Anonymous City, Anonymous Region, Anonymous Country. Correspondence to: Anonymous Author <anon.email@domain.com>.

Preliminary work. Under review by the International Conference on Machine Learning (ICML). Do not distribute.

not incorporate external contextual knowledge dynamically, making them less robust when faced with complex and evolving time series patterns. Furthermore, TSFMs often suffer from limited interpretability.

Recently, Retrieval-Augmented Generation (RAG) (Lewis et al., 2020) has demonstrated significant success across various Natural Language Processing (NLP) tasks by enhancing LLMs through the retrieval of relevant document segments from external knowledge bases. By incorporating retrieved information, RAG refines the existing prompts and generates more informed, context-aware outputs, improving both accuracy and adaptability in diverse applications. Inspired by this, in this paper, we present TS-RAG, a retrieval-augmented generation based time series forecasting framework to dynamically incorporate semantically relevant time series patterns into its forecasting pipeline, eliminating the need for fine-tuning while significantly improving zero-shot forecasting performance and the interpretability of TSFMs.

Instead of simply relying on the input time series query, TS-RAG first adopts pre-trained time series encoders to retrieve relevant time series segments from a dedicated knowledge database, providing valuable contextual knowledge for forecasting. Next, to effectively integrate retrieved time series knowledge, TS-RAG leverages a learnable Mixture-of-Experts (MoE)-based (Shazeer et al., 2017) augmentation module which can dynamically fuse retrieved patterns with the input time series query, ensuring that the model benefits from both existing knowledge and current query. With retrieval-augmented generation, TS-RAG not only can circumvent the need for fine-tuning on specific datasets but also can utilize retrieved segments to provide explicit rationales to explain the model's predictions. Finally, thorough empirical studies on seven public benchmark datasets demonstrate that TS-RAG achieves state-of-the-art zero-shot forecasting performance, outperforming existing TSFMs by up to 6.51% across diverse domains while simultaneously enhancing interpretability, reinforcing its potential as a robust and generalizable forecasting framework.

## 2. Related Work

### 2.1. Time Series Foundation Models

Recently, the rapid development of Time Series Foundation Models has drawn significant attention and made substantial progress in time series forecasting. These existing models, often adapted from advancements in Natural Language Processing (NLP) and Vision Transformers, have demonstrated strong generalization capabilities across diverse datasets. Lag-Llama (Rasul et al., 2023) and TimeGPT-1 (Garza & Mergenthaler-Canseco, 2023) are pioneering forecasting foundation models, pre-trained on extensive time series datasets spanning multiple domains. Lag-Llama uti-

lizes lagged time series features and the LLaMA architecture (Touvron et al., 2023a), while TimeGPT-1 adopts an encoder-decoder transformer structure to handle forecasting tasks effectively. Tiny Time Mixers (TTMs) (Ekambaram et al., 2024), built upon the earlier work TSMixer, trains a compact foundation model using multi-resolution data from various domains. TimesFM (Das et al., 2023) pretrains a patched-decoder attention model on a large time-series corpus to enable zero-shot forecasting across multiple domains. Chronos (Ansari et al., 2024) is an encoder-decoder style probabilistic time series foundation model which employs next-token-prediction for forecasting, enabling it to perform zero-shot forecasting on unseen forecast tasks. As a subsequent version of Chronos, Chronos-bolt incorporates a patch-based input strategy and uses decoder representations to generate quantile forecasts across multiple future steps, further improving forecast accuracy over its predecessor. Moirai (Woo et al., 2024) introduces a masked encoder-based universal time series forecasting transformer, accompanied by a novel large-scale time series dataset, LOSTA. Similarly, MOMENT (Goswami et al., 2024) compiles a large and diverse collection of public time series, called the Time series Pile, and systematically tackles the large-scale multi-dataset pretraining problem. These methods, however, lack inherent mechanisms to incorporate external contextual knowledge dynamically to facilitate zero-shot learning and suffer from limited interpretability.

### 2.2. Retrieval-Augmented for Time Series Forecasting

Existing LLM-based time series forecasters (Zhou et al., 2023; Jin et al., 2023; Pan et al., 2024) have demonstrated remarkable achievements in in-domain time series analysis. Nonetheless, adapting these models to different domains requires substantial computational resources. Given the challenges of adapting LLM-based models across different domains due to the computational costs, not only the previously mentioned time series foundation models offer a lightweight solution to this issue, but also the advent of Retrieval-Augmented Generation (RAG) techniques (Lewis et al., 2020) presents a compelling alternative. By augmenting the generative model's input with retrieved external knowledge, RAG has proven effective in open-domain question answering and document generation in the language field. Applying RAG to time series forecasting represents a novel and pioneering research direction. ReTime (Jing et al., 2022) proposes relational retrieval and content synthesis for spatial-temporal time series and time series imputation analysis. Yang et al. 2024 leverages K-means clustering to build a time series knowledge base and employs Dynamic Time Warping (DTW) as a similarity metric for retrieval. Liu et al. 2024 utilizes retrieved historical time series to guide the denoising process of diffusion model. However, existing works fail to unleash the full potential of RAG in

time series analysis. Our proposed TS-RAG is specially designed to enhance zero-shot forecasting in TSFMs, incorporating an innovative learnable Mixture-of-Experts (MoE) augmentation module that can dynamically fuse retrieved patterns with the input time series query.

# 3. TS-RAG for Zero-Shot Time Series Forecasting

**Overview**: The model architecture of the proposed TS-RAG consists of two key components, *i.e.*, retriever and augmentation, as shown in Figure 1. Given an input time series context window, a pretrained TSFM encoder first generates the context embedding. This embedding is then compared with time series context embeddings previously stored in the retrieval database to retrieve the top-$k$ similar time series pairs. Each retrieved pair includes a historical context and its corresponding forecasting horizon, which are utilized for augmentation and fusion to refine the zero-shot time series forecasting.

The retrieved future horizons of top–$k$ similar time series pairs are first transformed into embeddings and then fed into the Mixture-of-Experts (MoE) augmentation module along with the input time series embedding generated by the TSFM backbone. Note that we could select a different TSFM compared to the TSFM encoder in retriever. The MoE module adaptively assigns importance scores to these embeddings, dynamically integrating them into a unified representation. This final representation is then passed through the output projection layer of the TSFM to produce the enhanced time series forecast.

We provide detailed descriptions of the retrieval knowledge base, the TS-RAG framework, and the adaptive pretraining with zero-shot inference in the following sections.

## 3.1. Retrieval Knowledge Base for TS-RAG

TSFMs are typically pretrained on a multi-domain time series dataset, enabling them to perform zero-shot forecasting in unseen scenarios. Similarly, to enhance the generalization capability of TS-RAG, we construct a multi-domain dataset for its adaptive pretraining, specifically designed to learn the MoE module.

We leverage the pretraining dataset of Chronos (Ansari et al., 2024), which utilizes TSMixup to randomly combine time series data points from various domains. This approach enhances data diversity by blending different patterns, thereby improving the model's ability to generalize. Given that the additional parameters of TS-RAG are significantly less than those in the TSFM backbone, we sample a much smaller subset from the Chronos pretraining dataset to serve as the adaptive pretraining dataset for TS-RAG. From this smaller subset, we selectively extract a further subset to construct

the retrieval knowledge database for TS-RAG, which will be used in the forecasting phase.

The time series data stored in the knowledge base is processed into standard pairs, each consisting of a context window and its corresponding forecasting horizon. Formally, this can be expressed as: $\{(x_i, y_i) \,|\, i = 1, 2, \ldots, n\}$ where $x_i$ is the context window of the $i$-th time series, $y_i$ is the future horizon associated with $x_i$ and $n$ is the total number of pairs in the knowledge base.

Additionally, we employ a pretrained TSFM encoder to generate embeddings for the time series contexts stored in the retrieval knowledge base. These embeddings are then stored along with the corresponding time series data within the knowledge base. As a result, the structure of the retrieval database can be formally described as the following set of triplets:

$$\mathcal{D} = \{(x_i, e_i, y_i) \,|\, i = 1, 2, \ldots, n\}$$

where $\mathcal{D}$ denotes the retrieval database and $e_i$ denotes the embedding of $x_i$ generated by the pretrained TSFM encoder, resulting in a highly efficient training process.

## 3.2. Retrieve-Augmented Generation based Time Series Foundation Models

By leveraging relevant historical patterns retrieved from an external knowledge database, TS-RAG can enrich the query time series with additional contextual information, thereby improving both model generalization ability and prediction accuracy.

Within TS-RAG, a TSFM has three key components (Tan et al., 2024): an encoding layer, which may include normalization and embedding layers to preprocess and transform the input data; a backbone, typically implemented as a transformer based model (*i.e.* GPT (Radford et al., 2018), T5 (Raffel et al., 2020), Llama (Touvron et al., 2023b) *etc.*) to extract temporal representations; and a projection layer, often realized as a multi-layer perceptron (MLP), which maps the temporal representations from the backbone to the final prediction values.

TS-RAG introduces two additional components: a retriever and an augmentation module. These components work alongside the TSFM backbone, enabling the model to adaptively integrate retrieved information and improve forecasting accuracy. More specifically, the encoder from the pretrained Chronos model is used as the embedding model, which generates embeddings for both the query time series context and the contexts stored in the retrieval knowledge database. The retrieval process calculates the Euclidean distance between the query embedding and each stored context embedding in the knowledge base, and then selects the top-$k$ similar candidates based on the smallest distance.

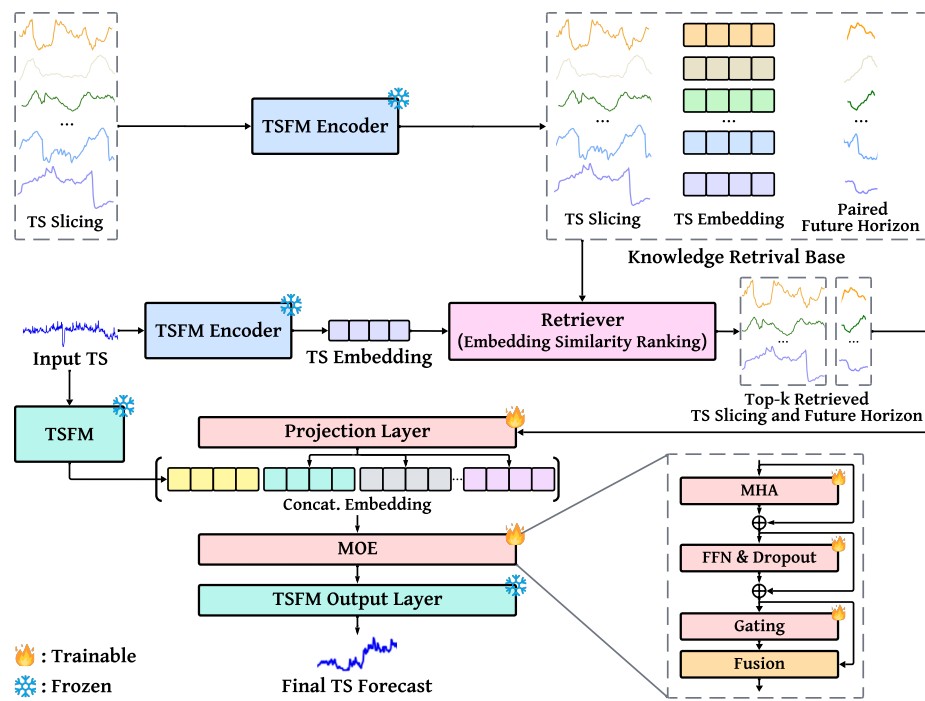

*Figure 1.* The TS-RAG model architecture processes an input time series by retrieving the top-$k$ similar time series and corresponding future horizons from a knowledge base through embedding similarity. These elements are fused with the input series embeddings using our MOE architecture to generate the final time series forecast.

Formally, given a query context $x_q$, we first compute its embedding using the Chronos encoder $f_{\text{encoder}}$:

$$e_q = f_{\text{encoder}}(x_q).$$

Next, the Euclidean distance between the query embedding and each stored embedding in the retrieval database is calculated:

$$d(e_q, e_i) = \|e_q - e_i\|_2, \quad \forall i \in \{1, 2, \ldots, n\}.$$

To identify the most relevant historical patterns, the retrieval mechanism selects the top-$k$ candidates with the smallest distances:

$$\mathcal{C} = \text{TopK}(\{(x_i, y_i, d(e_q, e_i)) \mid i = 1, 2, \ldots, n\}, k).$$

TopK($\cdot$) returns the top-$k$ entries ranked by the lowest distance values. The retrieved set $\mathcal{C}$ contains the most relevant context-forecast pairs, which are subsequently used to augment the forecasting process.

To perform forecasting, we develop a novel Mixture-of-Experts (MoE) augmentation module to integrate the projections of the top-$k$ retrieved forecasting horizons with the query time series embedding from the TSFM backbone to enhance prediction accuracy. Each embedding is treated as an expert, contributing to the final forecast. Initially, each

retrieved forecasting horizon $y_i$ is encoded independently using a learnable projection layer:

$$\hat{e}_i = f_{\text{MLP}}(y_i), \quad i = 1, 2, \ldots, k,$$

where $f_{\text{MLP}}$ is a feedforward network that maps each retrieved sequence into a dense representation. The resulting embeddings are stacked along a new dimension, forming:

$$E_{\text{enc}} = [\hat{e}_1, \hat{e}_2, \ldots, \hat{e}_k] \in \mathbb{R}^{k \times d}$$

where $d$ is the embedding dimension. To fuse the retrieved information with the query time series representation $\hat{e}_q \in \mathbb{R}^{1 \times d}$ generated by the TSFM, the two are concatenated into a single representation:

$$E_{\text{concat}} = [\hat{e}_q; E_{\text{enc}}] \in \mathbb{R}^{(k+1) \times d}.$$

This combined representation is passed through a Multi-Head Attention (MHA) layer to learn interactions between all the embeddings:

$$E_{\text{att}} = \text{MHA}(E_{\text{concat}})$$

where $E_{\text{att}} \in \mathbb{R}^{(k+1) \times d}$ represents the contextualized features. A gating mechanism is then applied to adaptively weight the contributions of the retrieved sequences and the model's original output. Specifically, a gating network computes scores for each expert:

$$\alpha = \text{Softmax}(W_g E_{\text{concat}} + b_g),$$

where $W_g$ and $b_g$ are learnable parameters, and $\alpha \in \mathbb{R}^{(k+1)\times 1}$ denotes the normalized attention weights. The fused representation is computed as a weighted sum while a skip connection is applied to preserve the TSFM's original ability:

$$e_{\text{final}} = \hat{e}_q + \sum_{i=1}^{k+1} \alpha_i E_{\text{att},i}.$$

Finally, the enriched sequence output $e_{\text{final}}$ is passed through the output projection layer of TSFM to generate the final forecast:

$$\hat{y}_q = f_{\text{proj}}(e_{\text{final}}),$$

This mechanism enhances the forecasting in several key aspects. By leveraging retrieved sequences, the model gains access to additional historical information, particularly valuable when the query context alone is insufficient for accurate predictions. The Multi-Head Attention mechanism enables the model to learn context-aware interactions between the retrieved data and its own predictions. Next, the gating mechanism adaptively determines the importance of each expert, allowing the model to focus on the most relevant information. Finally, the skip connection ensures that the model's initial predictions are preserved and enriched, maintaining a balance between internal knowledge and external augmentation. These properties collectively improve the prediction accuracy and enhance the interpretability of the model, particularly in zero-shot forecasting scenarios.

### 3.3. Adaptive Pretraining and Zero-shot Inference

During the adaptive pretraining phase, we selectively train only the external parameters of the MoE augmentation module in TS-RAG based on pre-constructed multi-domain datasets, while keeping all TSFM parameters frozen.

During zero-shot inference, TS-RAG utilizes its pretrained components to generate forecasts without any task-specific fine-tuning. The retrieval-augmented generation (RAG) approach enables TS-RAG to generalize across diverse forecasting tasks by leveraging external knowledge from a broad set of time series domains. By integrating relevant past trends, the model refines its predictions and mitigates uncertainties, leading to improved forecasting accuracy.

## 4. Experiments

### 4.1. Experimental Setup

**Datasets and Retrieval Knowledge Base**. For the pretraining dataset, we first sample 50 million data points from the Chronos pretraining dataset (Ansari et al., 2024) and further extract a subset of 5 million data points to construct the retrieval database. To facilitate efficient indexing and retrieval, both the pretraining dataset and the retrieval database are segmented using a predefined context window. This process results in a total of 26 million pretraining data pairs and 2.8 million retrieval database pairs.

The zero-shot experiments are conducted on widely recognized time series benchmark datasets spanning diverse domains, including ETTh1, ETTh2, ETTm1, ETTm2, Weather, Electricity, and Exchange Rate. Details of these datasets can be found in Appendix A.3. Zero-shot evaluation is performed on the test sets of these datasets, with a data split ratio of 6:2:2 for the ETT datasets and 7:1:2 for Weather, Electricity, and Exchange Rate.

During the zero-shot inference stage, the historical data of these datasets is expected to contain valuable patterns for augmentation. To utilize this, we build a retrieval database for each dataset using its own training set. This approach ensures that the retrieval mechanism captures relevant in-domain patterns, enhancing forecasting performance without fine-tuning any parameter.

**Baselines**. In practice, we use Chronos-Bolt, one of state-of-the-art TSFMs, as the backbone of TS-RAG, as it achieves competitive performance in our evaluations. While TS-RAG is designed to be compatible with any general TSFM, and we have verified its effectiveness beyond Chronos-Bolt, we primarily focus on this backbone in our experiments due to its strong empirical performance. For comparison, we also report the zero-shot performance of other TSFMs, including TTM (Ekambaram et al., 2024), TimesFM (Das et al., 2023), Moirai (Woo et al., 2024), Chronos (Ansari et al., 2024), Chronos-Bolt (Ansari et al., 2024), and MOMENT (Goswami et al., 2024).

**Setup**. Given that TSFMs are typically trained with a fixed forecasting length (*e.g.*, 64 or 96), we maintain this consistency in both pretraining and zero-shot evaluation. In our setup, the context length is set to 512, and the forecasting length is fixed at 64. We use Mean Squared Error (MSE) and Mean Absolute Error (MAE) as primary evaluation metrics to assess forecasting performance. The detailed definition of the evaluation metrics can be found in Appendix A.4.

### 4.2. Experimental Results for Zero-shot Forecasting

As shown in Table 1, TS-RAG$_{\text{Chronos-bolt}}$ consistently outperforms other TSFMs, including its backbone model, Chronos-Bolt, across all datasets. This demonstrates the effectiveness of RAG in leveraging relevant time series patterns from the external database to enhance zero-shot forecasting accuracy.

Compared to Chronos-Bolt, TS-RAG$_{\text{Chronos-bolt}}$ achieves an average reduction of 3.54% in MSE and 1.43% in MAE, confirming that the incorporation of retrieved information improves both precision and robustness. Notably, Chronos-Bolt already demonstrates strong performance on the ETTm1 dataset, achieving an MSE of 0.3109. However, TS-RAG$_{\text{Chronos-bolt}}$ further reduces MSE by 6.51%,

*Table 1.* Long-term zero-shot forecasting results. Best results are highlighted in **bold**, second best results are underlined. "—" indicates the datasets were used in pretraining and zero-shot results are not reported.

| Methods | TS-RAG$_{Chronos\text{-}bolt}$ | | Chronos-bolt$_B$ | | MOMENT | | TTM$_B$ | | Moirai$_B$ | | TimesFM | | Chronos$_B$ | |
|---|---|---|---|---|---|---|---|---|---|---|---|---|---|---|
| Metric | MSE | MAE | MSE | MAE | MSE | MAE | MSE | MAE | MSE | MAE | MSE | MAE | MSE | MAE |
| ETTh1 | **0.3557** | **0.3624** | 0.3616 | 0.3650 | 0.3920 | 0.4110 | 0.3619 | 0.3710 | 0.3686 | 0.3835 | 0.4254 | 0.3825 | 0.4217 | 0.3806 |
| ETTh2 | **0.2451** | **0.2982** | 0.2517 | 0.2992 | 0.2982 | 0.3585 | 0.2531 | 0.3032 | 0.2547 | 0.3053 | 0.2894 | 0.3233 | 0.2659 | 0.3136 |
| ETTm1 | **0.2906** | **0.3114** | 0.3109 | 0.3185 | 0.3506 | 0.3834 | 0.3152 | 0.3248 | 0.5399 | 0.4322 | 0.3321 | 0.3326 | 0.3935 | 0.3695 |
| ETTm2 | **0.1466** | **0.2231** | 0.1487 | 0.2236 | 0.1964 | 0.2847 | 0.1511 | 0.2405 | 0.1958 | 0.2687 | 0.1703 | 0.2552 | 0.1663 | 0.2522 |
| Weather | **0.1454** | **0.1771** | 0.1525 | 0.1825 | 0.1801 | 0.2384 | 0.1543 | 0.1893 | 0.1711 | 0.1912 | — | — | 0.1897 | 0.2107 |
| Electricity | **0.1120** | **0.2002** | 0.1132 | 0.2004 | 0.1967 | 0.3028 | 0.1715 | 0.2643 | 0.1832 | 0.2814 | — | — | 0.1460 | 0.2237 |
| Exchange rate | **0.0627** | **0.1718** | 0.0673 | 0.1780 | 0.0979 | 0.2059 | 0.0657 | 0.1725 | 0.0663 | 0.1720 | 0.0695 | 0.1802 | 0.0831 | 0.1879 |

highlighting its ability to refine forecasts even in scenarios where the backbone TSFM is already highly optimized.

Across individual datasets, TS-RAG$_{Chronos\text{-}bolt}$ consistently achieves the lowest MSE and MAE, demonstrating its robustness across diverse time series patterns. Significant performance gains are observed on datasets such as ETTm1 and Weather, where TS-RAG not only outperforms Chronos-Bolt but also surpasses all other TSFMs by a notable margin. This improvement suggests that RAG is particularly effective in datasets with complex temporal dependencies, where incorporating relevant time series patterns from existing database significantly enhances forecasting accuracy.

### 4.3. Ablation Studies

#### 4.3.1. SENSITIVITY TO THE NUMBER OF RETRIEVED SEQUENCES

The impact of varying the number of retrieved sequences ($k$) on forecasting performance is illustrated in Figure 2. The x-axis represents the number of retrieved sequences, while the y-axis shows the corresponding Mean Squared Error (MSE). Across all datasets, increasing $k$ initially leads to a significant decrease in MSE, demonstrating that incorporating additional retrieved sequences helps refine predictions by leveraging historical patterns. However, beyond a certain threshold, the improvement plateaus and even decreases slightly in some datasets, indicating diminishing returns as $k$ increases.

Dataset-specific trends further reveal differences in sensitivity to $k$. For instance, ETTm1 and ETTm2 exhibit the most pronounced improvement as $k$ increases, with MSE rapidly declining before stabilizing. This suggests that these datasets benefit significantly from retrieval-augmented learning, likely due to strong temporal dependencies in their historical patterns. ETTh1 and ETTh2 show a similar trend but with a smaller overall reduction in MSE, indicating that while retrieval is beneficial, these datasets may already con-

tain strong intrinsic signals, making additional augmentation less impactful. The Weather, Electricity, and Exchange Rate datasets display a steady decline in MSE with $k$ increasing, but beyond $k = 10$, the improvement becomes marginal, suggesting that a moderate number of retrieved sequences is sufficient.

While larger $k$ values generally improve performance, computational costs also increase. Based on the observed results, an optimal range of $k$ between 6 and 10 appears to provide a good trade-off between accuracy and efficiency. These findings suggest that TS-RAG benefits from a carefully chosen number of retrieved sequences rather than an indiscriminate increase.

#### 4.3.2. EFFECTIVENESS VS. DIFFERENT VERSIONS OF RETRIEVAL DATABASE

The impact of different versions of retrieval database choices on zero-shot forecasting performance is presented in Table 2. The results compare three settings: (1) w/o RAG, where no retrieval augmentation is applied, (2) pretrained database, where retrieval is performed using a database constructed from the Chronos pretraining dataset, and (3) historical database, where retrieval is performed using a dataset-specific retrieval database constructed from the training set of each dataset (only for inference, without fine-tuning).

Across two versions of retrieval databases, TS-RAG consistently outperforms the baseline w/o RAG setting, highlighting the effectiveness of retrieval-augmented generation. Between the two versions of retrieval databases, the one with historical database generally yields superior performance, achieving the lowest MSE in six out of seven datasets. This suggests that retrieving in-domain sequences from the same dataset provides more relevant contextual information than retrieving from a broader, pretraining-based retrieval set. However, on the ETTh2 dataset, TS-RAG achieves the best performance when using the pretrained database, implying that retrieval from a more diverse database may sometimes

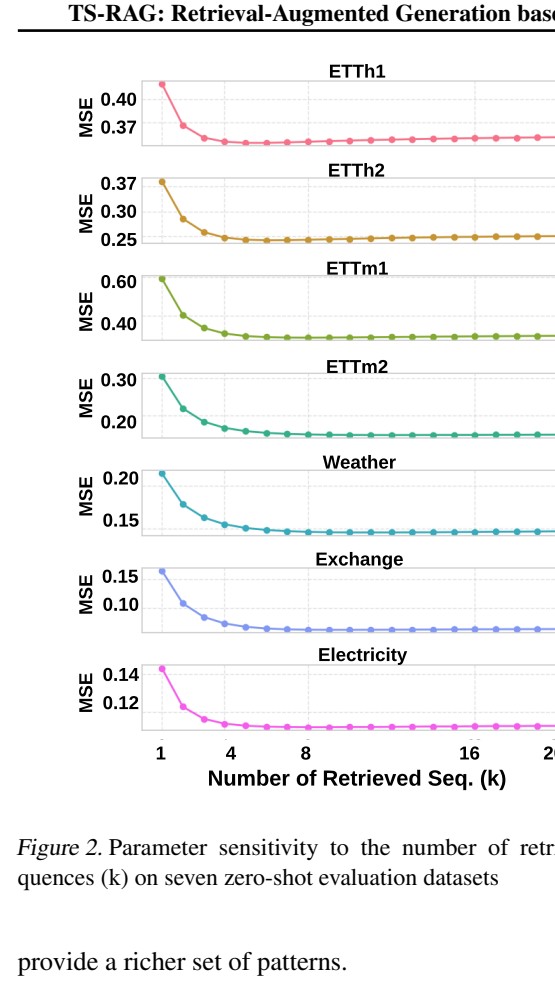

*Figure 2.* Parameter sensitivity to the number of retrieved sequences (k) on seven zero-shot evaluation datasets

provide a richer set of patterns.

The choice of retrieval database plays a crucial role in forecasting accuracy. While dataset-specific retrieval tends to be more effective, a multi-domain retrieval database can sometimes offer advantages, particularly in datasets with high variability. Future work could explore hybrid retrieval mechanisms to further enhance forecasting performance.

### 4.3.3. EFFECTIVENESS OF RETRIEVAL LOOKBACK LENGTHS

Table 3 presents the effect of different retrieval lookback lengths on zero-shot forecasting performance. Given an input sequence of length 512, we explore different retrieval configurations by using only the last 64, 128, or 256 time steps, or the full 512 time steps for retrieval.

Across all datasets, longer retrieval lookback windows (256 or 512) yield relatively better performance, suggesting that incorporating a more extended historical context helps retrieve more relevant sequences. This finding demonstrates that retrieving from longer historical sequences generally improves the quality of retrieved sequences, leading to greater forecasting accuracy. However, in some cases, longer is not always better, indicating that excessive retrieval windows may introduce noise or irrelevant information. This suggests the potential for adaptive retrieval mechanisms that allow

*Table 2.* Long-term zero-shot forecasting results with different retrieval database. Best results are highlighted in **bold**, and second-best results are underlined. MSE is reported here.

|  | w/o RAG | pretrain database | historical database |
|---|---|---|---|
| ETTh1 | 0.3616 | 0.3564 | **0.3557** |
| ETTh2 | 0.2517 | **0.2432** | 0.2451 |
| ETTm1 | 0.3109 | 0.2971 | **0.2906** |
| ETTm2 | 0.1487 | 0.1513 | **0.1466** |
| Weather | 0.1525 | 0.1502 | **0.1454** |
| Electricity | 0.1132 | 0.1125 | **0.1120** |
| Exchange rate | 0.0673 | 0.0639 | **0.0627** |

*Table 3.* Long-term zero-shot forecasting results with different retrieval lookback lengths. Best results are highlighted in **bold**, and second-best results are underlined. MSE is reported here.

|  | w/o RAG | 64 | 128 | 256 | 512 |
|---|---|---|---|---|---|
| ETTh1 | 0.3616 | 0.3540 | 0.3572 | **0.3539** | 0.3557 |
| ETTh2 | 0.2517 | 0.2432 | 0.2415 | **0.2409** | 0.2451 |
| ETTm1 | 0.3109 | 0.3114 | 0.2935 | 0.3195 | **0.2906** |
| ETTm2 | 0.1487 | 0.1502 | 0.1494 | 0.1518 | **0.1466** |
| Weather | 0.1525 | 0.1491 | 0.1526 | 0.1518 | **0.1454** |
| Electricity | 0.1132 | 0.1132 | 0.1125 | 0.1130 | **0.1120** |
| Exchange rate | 0.0673 | 0.0678 | 0.0674 | 0.0662 | **0.0627** |

the retriever to dynamically determine the most suitable retrieval lookback length for each instance.

### 4.3.4. EFFECTIVENESS ON LONGER FORECASTING HORIZONS

Time Series Foundation Models (TSFMs) typically employ a rolling strategy when forecasting a horizon longer than their pretraining length. In this approach, the model iteratively generates predictions for shorter segments and then rolls forward to forecast the next segment until the full horizon is covered. TS-RAG follows a similar strategy but enhances it with retrieval augmentation. Specifically, for each forecasting step, TS-RAG retrieves the next 64-step forecasting horizon from the retrieval database, incorporating relevant historical patterns at each iteration until the specified forecasting length is reached.

Table 4 presents the zero-shot forecasting results across multiple datasets, including ETTh1, ETTh2, Weather, and Exchange Rate. The results show that TS-RAG consistently outperforms its backbone model, demonstrating the effectiveness of retrieval-augmented generation (RAG) in extending prediction horizons while maintaining accuracy. The performance gain suggests that leveraging retrieved sequences mitigates error accumulation, a common issue in rolling-based forecasting.

*Table 4.* Zero-shot forecasting results for extended forecasting horizons across multiple datasets. We report MSE.

| Forecasting Length | 96 | | 192 | | 336 | | 720 | |
|---|---|---|---|---|---|---|---|---|
| Methods | w/o RAG | TS-RAG | w/o RAG | TS-RAG | w/o RAG | TS-RAG | w/o RAG | TS-RAG |
| ETTh1 | 0.3859 | **0.3781** | 0.4446 | **0.4343** | 0.4850 | **0.4679** | 0.4841 | **0.4602** |
| ETTh2 | 0.2899 | **0.2798** | 0.3603 | **0.3460** | 0.4045 | **0.3828** | 0.4143 | **0.3975** |
| Weather | 0.1777 | **0.1706** | 0.2244 | **0.2172** | 0.2838 | **0.2796** | 0.3673 | **0.3577** |
| Exchange rate | 0.0993 | **0.0942** | 0.1926 | **0.1819** | 0.3437 | **0.3129** | 0.8100 | **0.6701** |

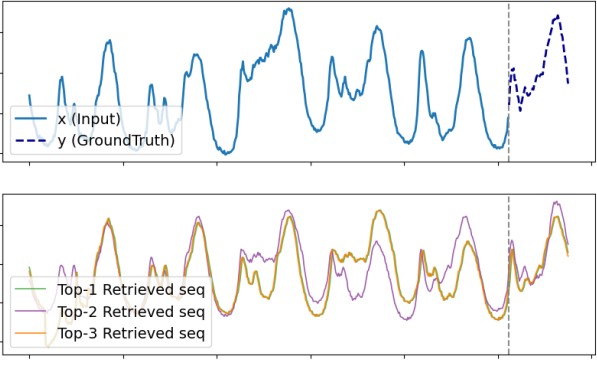

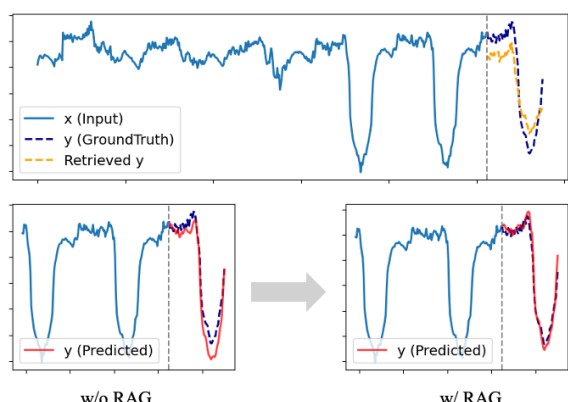

*Figure 3.* Case study on TS-RAG retrieval (Weather): Given the query time series, the retriever selects relevant historical sequences based on the embedding of the query. The retrieved sequences exhibit strong similarity to the input query in terms of both trend and periodicity.

*Figure 4.* Case study on TS-RAG retrieval and forecasting (ETTm1): Given the retrieved sequence, the forecasting result with RAG better aligns with the sharp downward trend.

### 4.4. Case Studies

To further illustrate the effectiveness of TS-RAG, we conduct case studies on retrieval quality and its impact on forecasting performance compared to the baseline TSFM (Chronos-bolt). As shown in Figure 3, the top panel presents the input query and corresponding forecasting horizon from the Weather dataset, while the bottom panel displays the retrieved sequences. The retriever selects the most relevant historical sequences based on the embedding similarity of query time series. The retrieved sequences exhibit strong alignment with the input in terms of both trend and periodicity, indicating that the retrieval process effectively captures complex temporal patterns. Moreover, the forecasting horizons associated with the retrieved sequences closely align with the real future horizon, demonstrating that retrieved-augmented forecasting provides valuable external information to refine and explain the forecasts.

Figure 4 showcases a case study from the ETTm1 dataset. The top panel illustrates the input query, its corresponding forecasting horizon, and the future horizon of a retrieved sequence, while the bottom panel compares the forecasting results of the backbone TSFM and TS-RAG. The TSFM without RAG fails to capture a sudden trend shift effectively. By retrieving similar patterns from the retrieval database, TS-RAG successfully adapts to the trend, leading to more accurate forecasts. More case studies are shown in Appendix B and demonstrate how retrieval-augmented

forecasting enhances robustness and interpretability across diverse real-world time series.

## 5. Conclusion

In this paper, we introduced TS-RAG, a novel retrieval-augmented forecasting framework designed to enhance the generalization and interpretability of Time Series Foundation Models (TSFMs) in zero-shot forecasting. By integrating retrieval-augmented generation (RAG) with a pre-trained TSFM encoder and a Mixture-of-Experts (MoE) based augmentation module, TS-RAG effectively incorporates retrieved relevant patterns to improve forecasting accuracy in previously unseen domains. Extensive empirical evaluations on multiple benchmark datasets demonstrate that TS-RAG consistently outperforms existing TSFMs, validating its effectiveness in zero-shot forecasting while maintaining strong interpretability.

Looking ahead, we aim to 1) explore multimodal extensions of TS-RAG by integrating heterogeneous time series data, such as text data, to further enhance forecasting capabilities; 2) we plan to investigate optimization techniques for retrieval ranking in RAG, assessing whether more effective retrieval mechanisms can further boost zero-shot forecasting performance. We believe that TS-RAG establishes a strong foundation for retrieval-augmented time series forecasting, setting up a new frontier for robust and adaptable time series forecasting in dynamic and open world environments.

## Impact Statement

This work enhances time series forecasting by leveraging RAG to improve time series foundation model performance. The broader impact of this work can be multifaceted. It may enhance decision-making in critical domains such as finance, healthcare, and environmental monitoring by providing more accurate and reliable forecasts and could lead to better resource allocation, improved patient care, and more effective responses to climate change. No ethical concerns must be considered.The social impacts are significant, as it has the potential to revolutionize our approach to complex time series data and the integration of emerging AI tools, including foundational models. It could change how we analyze and leverage time series data in various fields.

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

## A. Experimental Details

### A.1. Implementation Details

During pretraining, all parameters of the TSFM backbone are frozen, and only the additional parameters introduced by TS-RAG are fine-tuned. The number of retrieved sequences (top-$k$) is set to 10 by default; however, due to the flexible design of the MoE augmentation module, different values of $k$ can be explored. The model is trained using the AdamW optimizer with a learning rate of 0.0003 and a weight decay of 0.01. The batch size is set to 256, and training is conducted for 10,000 steps. To improve generalization, dropout is applied to certain layers with a dropout rate of 0.2. The training process was conducted on an NVIDIA A6000-48G GPU using TF32 precision. For efficient retrieval, FAISS is used to quickly identify the most relevant historical sequences from the retrieval database.

### A.2. Baseline Introduction

We introduce the baseline models that we choose to compare in the following section:

- **Chronos-bolt** (Ansari et al., 2024): Chronos-bolt is a subsequent version of Chronos, which can handle patch-based and uses decoder representations to generate quantile forecasts across multiple future steps, improving forecast accuracy over Chronos.

- **MOMENT** (Goswami et al., 2024): MOMENT uses a masking modeling technique for zero-shot forecasting by appending a lookback series with a mask that matches the length of the forecast. It involves pretraining a Transformer encoder model univariately on the "Time Series Pile" datasets, which includes a wide variety of time series data.

- **TTM** (Ekambaram et al., 2024): TTM pre-trains a compact model based on the light-weight TSMixer architecture. It incorporates adaptive patching, diverse resolution sampling, and resolution prefix tuning to pretrain successfully on a small dataset.

- **Moirai** (Woo et al., 2024): Moirai pretrains the Transformer encoder on the "LOTSA" dataset, which includes27B time points, by masking the forecast horizon of each target channel and performing mask reconstruction.

- **TimesFM** (Das et al., 2023): TimesFM employs a decoder-style attention model and is pre-trained in a univariate manner on a large group of both real-world and synthetic datasets.

- **Chronos** (Ansari et al., 2024): Chronos is a probabilistic time series foundation model.Chronos tokenizes the input time series into quantized manner and processes these tokens using the T5 model (Raffel et al., 2020). Chronos is trained on an extensive corpus of collected and synthetic time series data and has great generalization ability.

### A.3. Details of Inference Datasets

We experiment the zero-shot forecasting on the widely adopted Electricity Transformer Temperature (ETT) datasets (Zhou et al., 2021), Weather, Electricity (Wu et al., 2023), and Exchange Rate from (Lai et al., 2018). ETT datasets are comprised of roughly two years of data from two locations in China. The data are further divided into four distinct datasets, each with different sampling rates: ETTh1 and ETTh2 are sampled hourly, and ETTm1 and ETTm2 are sampled every 15 minutes. Every ETT dataset includes six power load features and a target variable: the oil temperature. The Electricity dataset comprises records of electricity consumption from 321 customers and is measured with a 1-hour sampling rate. The Weather dataset contains one-year records from 21 meteorological stations located in Germany. The sampling rate for the Weather dataset is 10 minutes. The Exchange Rate dataset includes the daily exchange rates of eight foreign countries, including Australia, British, Canada, Switzerland, China, Japan, New Zealand and Singapore ranging from 1990 to 2016.

### A.4. Evaluation Metrics

For evaluation metrics, we use the mean square error (MSE) and mean absolute error (MAE) for zero-shot forecasting. We present the calculations of these metrics as follows:

$$\text{MSE} = \frac{1}{H} \sum_{h=1}^{T} \left( \mathbf{Y}_h - \hat{\mathbf{Y}}_h \right)^2, \qquad \text{MAE} = \frac{1}{H} \sum_{h=1}^{H} \left| \mathbf{Y}_h - \hat{\mathbf{Y}}_h \right|$$

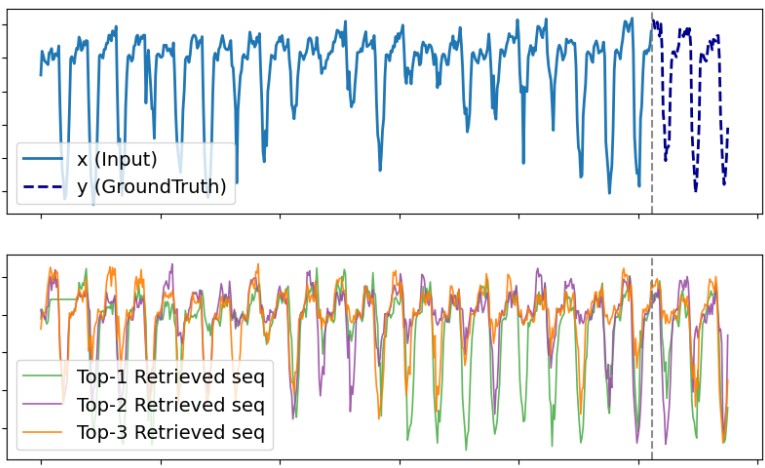

*Figure 5.* Retrieval results from the ETTh1 dataset.

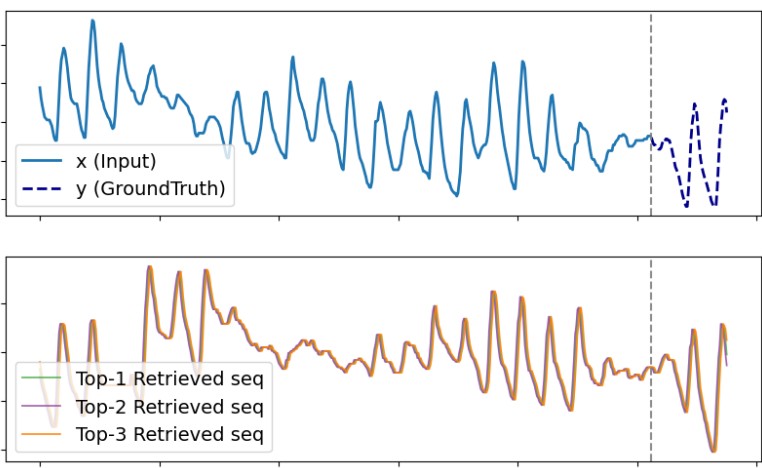

*Figure 6.* Retrieval results from the ETTh2 dataset.

where s is the time series data periodicity. H denotes the prediction intervals. $Y_h$ and $\hat{Y}_h$ are the $h$-th ground truth and prediction where h $\in \{1, ..., H\}$. For the evaluation metrics in long-term forecasting, we clarify that the reported metrics are the normalized versions of MAE/MSE. Although we apply global standardization to the data, the information that the scaler used is from training data solely.

## B. Showcases

### B.1. Case Studies on Retrieval Effectiveness

Figures 5 and 6 illustrate the retrieval performance of TS-RAG on the ETTh1 and ETTh2 datasets.The retrieval results demonstrate that TS-RAG effectively identifies historical patterns with strong structural similarity to the input, particularly in terms of periodicity and trend dynamics. In ETTh1, the retrieved sequences capture complex fluctuations and local variations, aligning well with the seasonal patterns of the input. Meanwhile, in ETTh2, where the time series exhibits smoother periodicity, the retrieved sequences show almost perfect alignment, indicating the presence of highly consistent cyclic behavior. These results suggest that retrieval augmentation enhances forecasting by leveraging historical patterns that closely match the current context, particularly in datasets with strong seasonal dependencies.

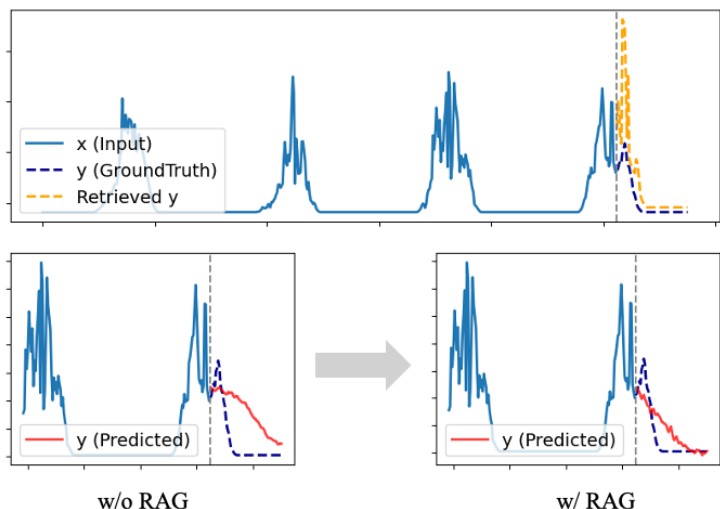

w/o RAG                    w/ RAG

*Figure 7.* Retrieval-Augment forecasting results from the Weather dataset–example of improving trend adaptation.

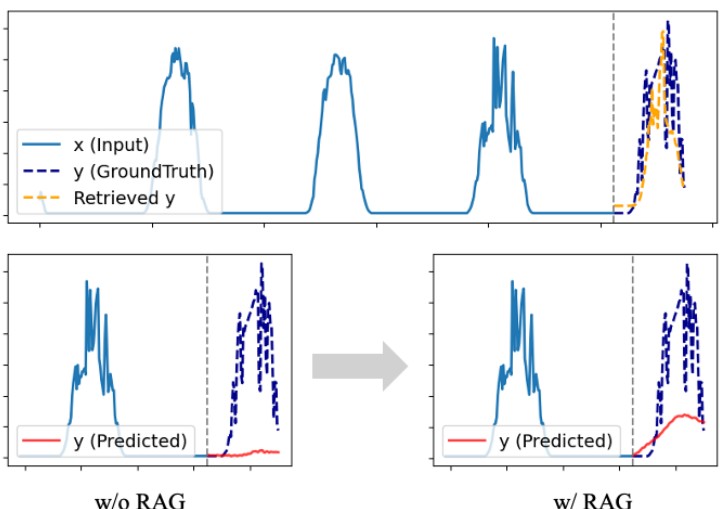

w/o RAG                    w/ RAG

*Figure 8.* Retrieval-Augment forecasting results from the Weather dataset–example of improving peak prediction.

## B.2. Case Studies on Retrieval-Augmented Forecasting

Figures 7 and 8 showcase the impact of retrieval augmentation on forecasting accuracy in the Weather dataset. Figure 7 highlights a situation that the baseline TSFM struggles to capture a sudden trend shift, leading to a significant forecasting error. By incorporating retrieved forecasting horizons, TS-RAG successfully adapts to the trend change. Figure 8 demonstrates how retrieval augmentation enhances peak prediction. The standard TSFM underestimates the upcoming peak, whereas TS-RAG, guided by similar retrieved patterns, generates a more accurate forecast. These case studies illustrate how retrieval-augmented forecasting helps models better adapt to complex temporal patterns, improving robustness in real-world forecasting tasks.

