# OpenReview forum: "TS-RAG: Retrieval-Augmented Generation based Time Series Foundation Models are Stronger Zero-Shot Forecaster"
_ICML.cc/2025/Conference — Submitted to ICML 2025_

### Official Review · Reviewer_4TxX · 2025-03-07

**Overall Recommendation:** 3

**Summary:**

This paper proposes TS-RAG, a retrieval-augmented forecasting framework that enhances zero-shot time series prediction by integrating retrieval-augmented generation (RAG) with a pretrained Time Series Foundation Model. The model consists of two key components: a retriever that selects relevant historical time series patterns from a retrieval knowledge base, and a Mixture-of-Experts (MoE) augmentation module that dynamically fuses retrieved sequences with the input query. TS-RAG leverages retrieved information to refine predictions, improving both accuracy and interpretability. The retrieval process is based on embedding similarity, identifying the most relevant sequences from a multi-domain database. The MoE module then adaptively assigns importance weights to retrieved sequences, ensuring effective knowledge integration.

**Claims And Evidence:**

Certain inconsistencies and ambiguities raise concerns about the clarity and validity of some claims in the paper.

C1: The text states that the TSFM encoder is pretrained to generate embeddings, while the figures suggest that the encoder is frozen during inference. This discrepancy affects the understanding of the retrieval process—if the encoder is frozen, it means no further domain adaptation occurs post-pretraining. However, if it is simply pretrained, it implies potential fine-tuning or adaptation.

C2: The paper claims that the augmentation module follows an MoE approach, but the actual implementation does not align with traditional MoE in Transformers or LLMs in my opinion. TS-RAG appears to simply apply a weighted sum over retrieved time series segments, without distinct expert specialization. This misalignment with conventional MoE terminology could be misleading and may require a more precise description.

C3: There is no clear mathematical formulation/ context description demonstrating the presence of self-attention/Transformer backbone mechanisms

**Essential References Not Discussed:**

[1] Tire K, Taga E O, Ildiz M E, et al. Retrieval Augmented Time Series Forecasting[J]. arXiv preprint arXiv:2411.08249, 2024.

**Experimental Designs Or Analyses:**

My concerns are same as 'Methods And Evaluation Criteria' Section.

**Methods And Evaluation Criteria:**

I have some concerns regarding the suitability of the benchmark datasets used.

The datasets used in the paper, including  Weather, ECL, exchage, and ETT, may not be ideal for evaluating long-term time series forecasting. For example, predicting weather for 30 days in advance might not be realistic from a physical standpoint. While these datasets are commonly used in previous methods like Informer, Autoformer, TimesNet, and TimeLLM, the concerns raised in the NeurIPS 2024 conference about the appropriateness of such datasets for long-term forecasting suggest that the relevance of these datasets should be reconsidered [1].

[1] Fundamental limitations of foundational forecasting models: The need for multimodality and rigorous evaluation, https://cbergmeir.com/talks/neurips2024/

**Other Comments Or Suggestions:**

Please refer to the question part.

**Other Strengths And Weaknesses:**

For strength, I think that the strength for this paper is:

* **Easy to Read**: The writing is clear and well-structured, and easy to read.

* **Innovative Use of RAG in Time Series Forecasting**: The paper extends RAG to time series forecasting, an underexplored yet promising direction.

* **Zero-Shot Forecasting Focus**: Addresses a critical challenge in time series modeling by improving generalization without fine-tuning, aligning with the trend of foundation models.

For weakness part, please refer to the question part.

**Questions For Authors:**

Q1 TO Q3: see **Claims And Evidence**

Q4:  The datasets used in the paper, including  Weather, ECL, exchage, and ETT, may not be ideal for evaluating long-term time series forecasting. For example, predicting weather for 30 days in advance might not be realistic from a physical standpoint. While these datasets are commonly used in previous methods like Informer, Autoformer, TimesNet, and TimeLLM, the concerns raised in the NeurIPS 2024 conference about the appropriateness of such datasets for long-term forecasting suggest that the relevance of these datasets should be reconsidered [1].

Q5: It appears that no supplementary material was provided, which raises concerns about reproducibility and methodological transparency. The lack of open-source code limits the ability of the community to replicate and validate the results. Without access to code or supplementary experiments, it becomes difficult to fully assess the robustness of the approach.

Q6:  Given that identical historical time series can lead to different futures (Time Series Data Phantom Issue), how does TS-RAG ensure that retrieval-based augmentation does not introduce misleading patterns? Additionally, since different types of time series may share similar historical patterns but contain different underlying information, how does TS-RAG prevent incorrect generalization across domains?

Q7:  The paper states: "Note that we could select a different TSFM compared to the TSFM encoder in retriever." What are the performances of using the same vs. a different backbone for retrieval and forecasting? Would using the same architecture improve compatibility between the retrieval and forecasting components?

Q8: Does TS-RAG always require a Transformer-based backbone?
If not, what alternative architectures could be used, and how would that affect forecasting performance?

Q9:  How are subsets selected from the Chronos pretraining dataset?
How is the retrieval knowledge base subset further extracted?

Q10:  How does the embedding method impact retrieval?
Different embedding techniques focus on different aspects of time series (e.g., trends, seasonality, local structure). How does TS-RAG ensure that the chosen embedding method aligns well with its retrieval goals?

Q11:  Should embeddings account for absolute timestamps (e.g., 12:00–18:00 vs. 02:00–08:00)? Two sequences might have similar embedding values but occur in completely different time contexts, potentially leading to incorrect retrieval. How does TS-RAG address this?

Q12: The derivation of  $\hat{e_q}$ is not explicitly explained.

Q13 The paper lacks formula numbering, which would drop clarity when referencing equations.

Q14 Standard Mixture-of-Experts (MoE) models often require load balancing to ensure different experts contribute meaningfully.
Does TS-RAG implement any mechanism to balance contributions across retrieved time series, or is the fusion process purely weight-based?

Q15: How is the 50M sample selection performed?
Is it ensured that the sampled dataset covers all categories of time series to avoid retrieval bias?

Q16:  If the model encounters completely new time series patterns, how does it avoid retrieving irrelevant sequences?
Does TS-RAG include a confidence measure or fallback mechanism for cases where retrieval provides non-useful sequences?

Q17: Real-world time series are not always of fixed length, unlike the fixed-length context windows used in this paper.
How does TS-RAG handle varying-length time series in embedding and retrieval queries?

Q18:  Compared to other models, TS-RAG effectively leverages more pre-existing information.
Could this additional prior knowledge be the primary reason for its superior performance, rather than the retrieval mechanism itself?

Q19: The paper states that increasing $K$ improves performance but increases computational cost, but no theoretical or experimental analysis is provided.

Q20: Prior research [2] suggests that introducing noisy or less relevant documents in RAG does not always degrade performance and can sometimes improve it.
What would happen if TS-RAG introduced less relevant retrieved sequences? Would this improve robustness or degrade accuracy?

Q21 What is the proportion of retrieving same-domain time series (e.g., predicting traffic using only traffic data) vs. cross-domain time series (e.g., predicting traffic using electricity demand patterns)?
Does TS-RAG favor in-domain retrieval, or does cross-domain retrieval provide additional generalization benefits?

[1] Fundamental limitations of foundational forecasting models: The need for multimodality and rigorous evaluation, https://cbergmeir.com/talks/neurips2024/

[2]  The Power of Noise: Redefining Retrieval for RAG Systems

**Relation To Broader Scientific Literature:**

The use of RAG for time series forecasting is a promising and emerging idea. While RAG has been widely explored in NLP, its application in time series forecasting remains relatively underexplored, with only a few recent works attempting to integrate retrieval-based enhancements into temporal modeling.

This paper contributes to this growing area by proposing TS-RAG, which aims to leverage retrieval-augmented learning for zero-shot forecasting. Although there have been some prior studies applying retrieval mechanisms in time series tasks (e.g., ReTime for relational retrieval in spatiotemporal data, or retrieval-based knowledge augmentation for diffusion models in time series), this work extends the idea.

**Theoretical Claims:**

The paper does not include formal mathematical proofs or rigorous theoretical claims that would require verification.

---

> ### Author Rebuttal · Authors · 2025-04-01
>
> Q1:
> Thank you for your concern. The TSFM encoder used for retrieval is frozen during both pretraining and inference. It is directly adapted from a pretrained TSFM model on diverse datasets, requiring no further fine-tuning.
>
> Q2:
> Thank you for the observation. We agree our augmentation module differs from standard MoE and is better described as a retrieval-based soft MoE [1]. Retrieved horizon representations act as experts, with a gating mechanism assigning soft weights to all top-k segments—unlike traditional MoE, which uses top-1 or top-2 for sparsity. We’ll revise the description accordingly in the updated paper.
>
> Q3: We will add the formulation of the transformer in the updated version.
>
> Q4: We appreciate the reviewer’s concerns and acknowledge the limitations of current benchmark datasets. We use these datasets as they are widely recognized and offer a standardized testbed for fair comparisons. That said, we agree that more realistic, diverse and especially multimodal benchmarks are necessary for better reflecting real-world forecasting challenges.
>
> Q5: The codes can be found here: https://shorturl.at/l1Fv4
>
> Q6: To avoid misleading retrievals, the retriever encoder in TS-RAG is pretrained on a forecasting task using future values as supervision, ensuring that series with similar future dynamics are mapped to similar embeddings. Pretraining on diverse, multi-domain datasets further enables the model to capture various temporal patterns and domain-specific features. The MoE module further helps avoid the negative impact of misleading patterns (See answer to Q2, reviewer 4x4W).
>
> Q7& Q10: Please refer to the response of Q6, and answer to Q4 for reviewer 4x4W
>
> Q8: TS-RAG is not limited to Transformer-based TSFMs; it can be built upon any architecture. The forecasting performance largely depends on the representation quality of the chosen backbone. Since TS-RAG aims to enhance the base model, stronger TSFMs generally yield better results. We also provide a MOMENT-based TS-RAG result in the anonymous repo.
>
> Q9: We randomly sample subset from the full Chronos pretraining dataset. The retrieval knowledge base is then randomly selected from the pretraining subset. Note that for retrieval, we remove sequences with a Euclidean distance of zero.
>
> Q11: We observed that top-k retrieved series often share similar timestamps with the query, suggesting temporal information is encoded in the embeddings. As noted in Q6, similar embeddings imply similar future dynamics. Even with mismatched time contexts, the MoE module adaptively weights and fuses useful patterns to improve forecasting.
>
> Q12: We mentioned in line 202 “the query time series representation eˆq ∈ R 1×d generated by the TSFM"
>
> Q13: We will include the formula numbering in the updated version.
>
> Q14: TS-RAG does not implement a load-balancing mechanism; the fusion process is purely weight-based.
>
> Q15: Refer to Q9
>
> Q16: First, the pretrained retriever encoder helps retrieve series with similar future dynamics (see Q6). Second, we use domain-aligned historical data to reduce irrelevant retrievals. Third, since retrieved horizons may differ from targets, TS-RAG dynamically integrate retrieved contexts for better predictions (see Q11). Finally, we appreciate the suggestion that adding confidence or fallback mechanisms is a valuable future direction.
>
> Q17: Similar to NLP, TS-RAG handles variable-length series using padding to standardize inputs, enabling comparison in a shared embedding space. As a future work, we plan to explore variable-length knowledge bases with adaptive retrieval tailored to each series’ natural length.
>
> Q18: We would like to emphasize that TS-RAG is a general framework designed to enable existing TSFMs to effectively utilize external knowledge. A stronger TSFM backbone naturally contributes to better performance. But the retrieval mechanism also plays a critical role by providing additional gains in accuracy and interpretability.
>
> Q19:With K kept small (≤20), the overhead is minor—e.g., at K = 5/10/15, forward pass takes 0.36/0.44/0.54 ms per query—showing limited impact in practice. A more detailed analysis will be included in the updated paper.
>
> Q20: Thank you for highlighting this paper. While RAG systems in QA are sensitive to noisy documents, the impact of noise in TS-RAG remains unclear. We agree that it's a valuable future direction and will explore it further in subsequent studies.
>
> Q21: Thank you for the thoughtful question. We conducted ablation studies under two settings: (1) distribution shift (e.g., using ETTh1 as knowledge base for ETTh2) and (2) cross-domain retrieval (e.g., using weather data as knowledge base for ETT). The results (https://shorturl.at/JxB7x) show TS-RAG performs best with same-domain retrieval and worst in cross-domain, highlighting the importance of domain alignment.
>
> **We will add the missed reference in the updated version.**
>
> [1] Puigcerver, Joan, et al. "From sparse to soft mixtures of experts."

---

> > ### Comment · Reviewer_4TxX · 2025-04-02
> >
> > Thank you for the response, which has resolved most of my concerns. Although I still harbor doubts about whether the Weather, ECL, Exchange, and ETT datasets are genuinely suitable for long-horizon forecasting, I acknowledge that this issue extends beyond the scope of this paper. The authors' explanation—that these datasets serve as community-recognized, widely used benchmarks facilitating fair comparison and reproducibility—is acceptable. Accordingly, I have decided to raise my score to 3.

---

> > > ### Author Response · Authors · 2025-04-05
> > >
> > > We sincerely appreciate the reviewer’s constructive comments and thoughtful feedback. We're glad that our rebuttal helped address your concerns and are grateful for the updated score and support. Your insights have been valuable in improving the clarity and quality of the paper. Thank you again for your time and careful review.

---

### Official Review · Reviewer_uqbX · 2025-03-13

**Overall Recommendation:** 3

**Summary:**

The authors introduce TS-RAG, a method designed to enhance the performance of a Time-Series Foundation Model (TSFM) by augmenting time-series sequences using an external database. The approach leverages the TSFM’s encoder to embed the input query, retrieve relevant candidate sequences, and weight them using a Mixture of Experts (MOE) module before merging them with the original query for final output generation. Experimental results validate the effectiveness of the proposed method.

## update after rebuttal
Thank you for responding to my questions. For now, I will keep the score as is.
## reviewer's after-rebuttal response ends here

**Claims And Evidence:**

Yes

**Essential References Not Discussed:**

NA

**Experimental Designs Or Analyses:**

Yes.

**Methods And Evaluation Criteria:**

Yes

**Other Comments Or Suggestions:**

The authors could consider sharing the implementation code via an anonymous link for reproducibility.

**Other Strengths And Weaknesses:**

Strengths:

1. TS-RAG effectively utilizes the TSFM’s embeddings to retrieve relevant sequences from an external database, presenting an innovative way to enhance time-series forecasting. The application of MOE is well-motivated and appropriate.
2. The evaluation is conducted on widely used benchmark datasets and state-of-the-art TSFMs, ensuring a rigorous validation of the proposed method.

Weaknesses:
1. The Abstract and Introduction claim that the proposed method enhances interpretability, but this aspect is not revisited or substantiated in the later sections of the paper.
2. The effectiveness of TS-RAG is demonstrated primarily using Chronos. Including results from additional TSFMs would provide a more comprehensive understanding of its generalizability.

**Questions For Authors:**

1. In the formulation of $e_{final}$, each component of $E_{att}$ corresponds to the forecast horizon, whereas $\hat{e_q}$ represents the input query. Since they correspond to different regimes of the input time series, what is the rationale behind combining them?
2. How is the data split ratio determined for the ETT and Weather datasets?
3. The pre-training database is a subset of the Chronos training data. What is the reasoning behind this choice? Would using a different dataset for augmentation be more beneficial in introducing unseen patterns to the model?

**Relation To Broader Scientific Literature:**

There is a growing literature on the RAG systems and Time series foundation models. There is very little work on understanding the applicability of RAG in TIme series foundation models. This paper attempts to apply a RAG framework on TSFMs.

**Theoretical Claims:**

There are no theoretical claims.

---

> ### Author Rebuttal · Authors · 2025-04-01
>
> **Reproducibility:** Thank you for your suggestion, we are glad to provide code, pretrained models, datasets and knowledge base via the anonymous link: *https://anonymous.4open.science/r/TS-RAG-F4DB*
>
> **W1: TS-RAG enhances interpretability**  Thank you for your comment. TS-RAG improves interpretability in two key ways: Compared to traditional TSFMs, which often act as black-box models, TS-RAG introduces a retrieval mechanism that explicitly provides similar historical sequences. This allows users to visually examine the retrieved sequences and understand how the model’s forecast is influenced by past events and patterns. During the retrieval and augmentation process, similarity scores and weights are computed, which can be used to highlight the most relevant historical patterns. This provides users with insight into which particular part of historical data contributes most to the prediction, helping them focus on the most informative patterns. We will provide a more comprehensive description of interpretability and show case studies in the updated version of the paper.
>
> **W2: TS-RAG on other backbone**  To better evaluate TS-RAG as a general framework, we also implement TS-RAG using MOMENT as the backbone, which consistently outperforms the original MOMENT. The results are in *https://anonymous.4open.science/r/TS-RAG-F4DB/Rebuttal%20and%20Discussion.md* **TS-RAG on other TSFM backbones**, which provide strong evidence for the effectiveness of the TS-RAG framework.
>
> **Q1 Rationale to combine E_att and e_q^:**  Thank you for your question. I'd like to clarify that E_att represents the combined embeddings of the query time series and the retrieved future horizons after being processed by an MHA layer. It contains k + 1 components: one for the query and k for the retrieved sequences.  The eq^ is the representation of the input query, in the TSFM forward process without RAG, eq^ is passed through a prediction head to generate the forecast. In our approach, we introduce a projection layer to map the retrieved forecast horizons into the same representation space as eq^, this alignment enables effective fusion of the query representation and the retrieved information, leading to forecasting performance improvements.
>
> **Q2 How to determine data split ratio:**  We follow the standard data split convention used in published papers[1,2].
>
> Ref:
>
> [1] Wu, Haixu, et al. "Timesnet: Temporal 2d-variation modeling for general time series analysis." arXiv preprint arXiv:2210.02186 (2022).
>
> [2] Jin, Ming, et al. "Time-llm: Time series forecasting by reprogramming large language models." arXiv preprint arXiv:2310.01728 (2023).
>
>  **Q3 Rational choosing subset of the Chronos training data:** The pre-training database is a subset of Chronos’s training data.  1) This is a consideration of the trade-off between model performance and retrieval efficiency. 2) This choice ensures that Chronos-bolt is already familiar with the training data, as we will freeze the parameters of Chronos-bolt and only train the MoE module’s augmentation mechanism. 3) The Chronos’s training data already contains diverse types of time series. Using a different dataset for post-training the TSFM backbone could improve generalization on the targeted specific dataset, but may affect its performance over the original dataset/generalization to other unseen data if the new dataset is not large enough and as diverse as the original dataset.

---

### Official Review · Reviewer_3i57 · 2025-03-13

**Overall Recommendation:** 4

**Summary:**

This paper proposes TS-RAG, a retrieval-augmented generation based method aimed at improving the generalization ability and interpretability of time series forecasting tasks. This framework does not require task-specific fine-tuning, enabling effective zero-shot forecasting while also providing interpretability for the predictions.

**Claims And Evidence:**

Yes

**Essential References Not Discussed:**

No

**Experimental Designs Or Analyses:**

I have checked the experimental setup, the analysis of the experiments, and the ablation studies. All the designs and analyses are reasonable and valid.

**Methods And Evaluation Criteria:**

Yes

**Other Comments Or Suggestions:**

The contributions can be summarized at the end of the introduction to enhance readability.

**Other Strengths And Weaknesses:**

Strengths:
The authors explore how to leverage the features extracted by foundation models and the rich information retrieved from dedicated databases, offering a promising solution for time series forecasting tasks. The paper is well-written and supported by a comprehensive set of experiments.

Weakness:
1.	Which loss function is used in training? More description is needed to reproduce the results.
2.	The authors compared the performance under different foundation models. A comparison with related works should also be provided for reference.

**Questions For Authors:**

Based on experience, the top-k samples retrieved from the dataset are crucial for the forecasting of the current time series. In other words, the model’s generalization ability can be attributed to the powerful backbone and the rich retrieval information. How can we know the effectiveness of the Mixture-of-Experts module?

**Relation To Broader Scientific Literature:**

Many studies have demonstrated that developing foundation models for time series tasks can effectively handle complex temporal dynamics. Therefore, leveraging the semantic information provided by these foundation models is highly valuable for time series tasks.

**Theoretical Claims:**

I have checked the derivation of the formulas.

---

> ### Author Rebuttal · Authors · 2025-04-01
>
> **W1: Training loss and Reproduce** Thank you for your question. TS-RAG uses the same loss as the backbone TSFM during training. Specifically, when using Chronos-bolt as the backbone, we adopt the quantile regression loss used in its original implementation.  For full reproducibility, we are glad to provide code, pretrained models, datasets and knowledge base via the link: *https://anonymous.4open.science/r/TS-RAG-F4DB*.
>
> **W2: TS-RAG on other backbone** Thank you for your suggestion. We agree that comparing with related works would provide valuable points. However, to the best of our knowledge, existing related works that combine retrieval-augmented forecasting with zero-shot evaluation do not publicly release their code and models. That said, we have ensured fair comparisons between various TSFMs and TS-RAG with different backbone TSFMs. Our experiments with TS-RAG applied to Chronos-bolt and MOMENT demonstrate that it can significantly improve the performance of the base TSFMs, highlighting the effectiveness of TS-RAG. We will continue to monitor future releases to include such open-source baselines.
>
> **Q1: Effectiveness of the MoE** Thank you for your insightful comment. In the TS-RAG system, the augmentation module is as important as the retriever, as it determines how the retrieved information is integrated into the final prediction. An ineffective fusion mechanism can lead to poor performance.  To demonstrate the importance of the Mixture-of-Experts module in TS-RAG, we compare it with a simpler alternative: a gated fusion module that linearly combines the original forecast with the retrieved forecasting horizon (similar to the fusion module in Time-MMD [1]).  And we conduct experiments with both augmentation modules under the same pretrain-zeroshot setting. The results in  *https://anonymous.4open.science/r/TS-RAG-F4DB/Rebuttal%20and%20Discussion.md* **Effectiveness of Mixture-of-Experts** show that although TS-RAG with a gated fusion module also improves the zero-shot performance, the performance gains are consistently lower than TS-RAG with the MoE module. This provides strong evidence of the effectiveness of the MoE module.
>
>
> Ref:
> [1] Liu, Haoxin, et al. "Time-MMD: Multi-Domain Multimodal Dataset for Time Series Analysis." NeurIPS Datasets and Benchmarks Track (2024).

---

### Official Review · Reviewer_4x4W · 2025-03-14

**Overall Recommendation:** 2

**Summary:**

This paper presents TS-RAG, a retrieval-augmented-generation-based time series forecasting framework. TS-RAG leverages pre-trained time series encoders to retrieve semantically relevant time series segments from a dedicated knowledge database. Next, it develops a learnable Mixture-of-Experts (MoE)-based augmentation module, which dynamically fuses retrieved time series patterns with the TSFM’s representation of the input query, improving forecasting accuracy without requiring task-specific fine-tuning. This paper evaluates TS-RAG on seven public benchmark datasets, demonstrating that TS-RAG achieves state-of-the-art zero-shot forecasting performance.

## update after rebuttal

I think this paper's main contribution lies in exploring the use of RAG for time series foundation models with some designs and experiments.  Some limitations are:
1. How the data in the knowledge base influences the RAG needs further exploration and discussion. When there are in-domain data, this method's relationship with fine-tuning needs to be discussed (such as comparing their performance and efficiency quantitatively, showing their advantages at different aspects). When there are no in-domain data, whether this method still works lacks some rigid guarantees.
2. The designed method is a direct implementation of RAG into time series models, which does not show many novel designs. The improvements are not clear enough on some models, such as Chronos, and the additional cost introduced by RAG is relatively high compared with simple zero-shot inference.

**Claims And Evidence:**

This paper claims that Time Series Foundation Models (TSFMs) lack inherent mechanisms for domain adaptation and are less robust when faced with complex and evolving time series patterns. It is not supported by convincing evidence. It is also unclear by comparing TSFMs with which other models can we get this conclusion.

**Essential References Not Discussed:**

There are no essential references not discussed.

**Experimental Designs Or Analyses:**

1) From Table 1, the performance improvement is not significant compared with existing models such as Chronos. Some other SOTA TSFMs are missing, such as Time-MoE [1].

2) It seems that this paper only combines TS-RAG with Chronos and does not show its performance on other TSFM backbones.

3) Table 2 discusses using the historical database. However, in this case, the database and the queries are from the same dataset, which cannot be considered as zero-shot forecasting.

[1] Time-moe: Billion-scale time series foundation models with mixture of experts

**Methods And Evaluation Criteria:**

The proposed method does not make sense to me.

1) It seems that TS-RAG highly depends on the high similarity between time series in the knowledge base and the query. How can we guarantee this when performing zero-shot forecasting? Are there any failure cases of this method in situations where no similar knowledge can be retrieved?

2) The retrieval is based on the Euclidean distances between series. Do time series with small Euclidean distances necessarily share similar dynamics or future horizons?

3) It is unclear how to select the encoder for the Retriever. How would this selection affect the performance?

4) It is confusing why we should use subsets to train TS-RAG and serve as the knowledge base since more data may lead to better performance.

**Other Comments Or Suggestions:**

No other comments.

**Other Strengths And Weaknesses:**

Other strengths:

1) The proposed method is easy to follow.

2) It is an interesting topic to consider enhancing TSFMs with additional knowledge bases.

Other weaknesses:

1) There needs to be more descriptions on how TS-RAG improves interpretability.

2) It would be better to discuss the training and inference costs of TS-RAG.

**Questions For Authors:**

Authors are suggested to address the issues or concerns mentioned in the above sections considering claims, methods, and experiments.

**Relation To Broader Scientific Literature:**

This paper tries to improve the performance of existing TSFMs with RAG techniques. The performance improvements are not large compared with TSFMs. The ideas of RAG and MoE in the proposed method are commonly used in time series forecasting or deep learning, and this paper does not make significant changes to these original ideas.

**Theoretical Claims:**

There is no proof for theoretical claims.

---

> ### Author Rebuttal · Authors · 2025-04-01
>
> **Q1: Claims**
>
> We appreciate the reviewer’s concern. While TSFMs are pretrained on diverse datasets and perform well in zero-shot and few-shot settings, they can still struggle with non-stationary data and distribution shifts. Existing TSFMs lack mechanisms to deal with this problem, which motivates our work. TS-RAG addresses this gap by introducing retrieval-based augmentation to enhance adaptability. We will revise the manuscript to clarify this point without overstating the limitations of current TSFMs.
>
> **Q2: How to guarantee high similarity**
>
> Thank you for your question. TS-RAG uses two mechanisms to ensure effective retrieval: 1. building the knowledge base from in-domain data, and 2. using a pretrained retriever encoder that captures future dynamics. Even if no highly similar sequences are included in the knowledge base, the MoE module can adaptively weight and fuse the retrieved patterns, ensuring the performance does not fall below the backbone TSFM.
>
> **Q3: Euclidean distances effectiveness**
>
> Thank you for your comments. To clarify, the Euclidean distance is calculated not in the raw time series space, but in the embedding space generated by a pretrained encoder. During pretraining for the encoder, it's optimized via backpropagation using future values as supervision, so that the embeddings of input sequences are well aligned with their future horizons. As a result, sequences with similar embeddings tend to share similar future dynamics.
>
> **Q4: Encoder choice**
>
> Thank you for your question. The retriever encoder should be pretrained on a forecasting task, while its architecture does not matter. Section 5.3 of [1] also shows that retrieval encoder choice has minimal effect on performance.
>
> **Q5: Subsets for training**
>
> Thank you for your question. The use of subsets for both training and the knowledge base is motivated by a trade-off between the model performance and retrieval efficiency.
>
> **Q6: Significance and Time-MoE**
>
> Thank you for the comments. When applied to another backbone (MOMENT), TS-RAG achieves 11.2% average and 21% max MSE improvement, showing that our method brings consistent and significant gains over different backbones.
> We also provide the results of Time-MoE. The tables can be found in the anonymous library https://shorturl.at/JxB7x.
>
> **Q7: Other backbones**
>
> Please refer to answer to **Q6**
>
> **Q8: Zero-shot setting**
>
> Thank you for your comments. We understand the concern for the zero-shot setting and would like to clarify this. The models used for embedding and forecasting are not trained on the target dataset. Our model is pretrained on a different dataset and directly applied to the new domain without additional fine-tuning. This aligns with the classical definition of zero-shot learning in literature [2,3,4].
>
> **Q9: Interpretability**
>
> Thank you for your comment. TS-RAG improves interpretability in two key ways:
> 1. Compared to traditional TSFMs, which often act as black-box models, TS-RAG introduces a retrieval mechanism that explicitly provides similar historical sequences. This allows users to visually examine the retrieved sequences and understand how the model’s forecast is influenced by past events and patterns.
> 2. During the retrieval and augmentation process, similarity scores and weights are computed, which can be used to highlight the most relevant historical patterns. This provides users with insight into which particular part of historical data contributes most to the prediction, helping them to focus on the most informative patterns.
>
> **Q10: Training and inference costs**
>
> Thank you for your suggestion.
> 1. Training: TS-RAG maintains efficiency by freezing the TSFM backbone and only training additional parameters. Preprocessing and caching retrieval indices further optimize training, taking approximately 1 hour on a single NVIDIA A6000 GPU with Chronos-bolt.
> 2. Inference: During inference, the additional cost mainly comes from the retrieval. TS-RAG uses Faiss for vector similarity search. It retrieves the top-k most similar sequences from the knowledge base. On the ETTh1 dataset, the retrieval process adds 9.2 ms of latency per query, the forward process adds another 0.44 ms. In total, the inference takes 9.62 ms per query, which remains practical for real-time applications.
>
> [1] Liu, Jingwei, et al. "Retrieval-augmented diffusion models for time series forecasting." Advances in Neural Information Processing Systems 37 (2024): 2766-2786.
>
> [2] Xian, Yongqin, et al. "Zero-shot learning — A comprehensive evaluation of the good, the bad and the ugly." Proceedings of the IEEE Conference on Computer Vision and Pattern Recognition (CVPR). 2018.
>
> [3] Brown, Tom, et al. "Language models are few-shot learners." Advances in Neural Information Processing Systems 33 (2020): 1877–1901.
>
> [4] Das, Abhimanyu, et al. "A Decoder-Only Foundation Model for Time-Series Forecasting." arXiv preprint arXiv:2310.10688 (2023).

---

> > ### Comment · Reviewer_4x4W · 2025-04-03
> >
> > Thank the authors for the rebuttal, which addresses some of my concerns. However, there are some remaining ones:
> >
> > 1. About in-domain data: One advantage of the TSFMs is their zero-shot capability, which handles out-of-distribution or new domains without enough data. The need for in-domain data limits model capabilities in such scenarios. Furthermore,  if there exist in-domain data, training a new model or fine-tuning the TSFMs may be more straightforward solutions.
> >
> > 2. About the pretrained encoder: It is not convincing to claim that the choice of the encoder does not make a difference since models pretrained with different architectures and pretraining data may have different performance, which will influence their measurement of data similarities.
> >
> > 3. It remains unclear how the subset size controls the trade-off between the model performance and retrieval efficiency, e.g., what are the performance and efficiency when we choose different subset sizes?
> >
> > 4. It seems that the original MOMENT model does not use zero-shot inference in long-horizon forecasting, and thus, using TS-RAG on this model may not be convincing.
> >
> > 5. During inference, it seems that the latency caused by retrieval is relatively high compared with the forward process.

---

> > > ### Author Response · Authors · 2025-04-05
> > >
> > > **We sincerely thank the reviewer for the continued engagement and thoughtful feedback. We're pleased that our previous response helped address your concerns, and we appreciate the opportunity to further clarify the remaining details.**
> > >
> > > **1. About in-domain data and fine-tuning**
> > >
> > > Thank you for the comment.
> > >
> > > (1) **Effectiveness without in-domain data:** We would like to clarify that TS-RAG is still effective even without in-domain data. But if in-domain data is available, the performance of TS-RAG can be further improved. Specifically, the experiments¹ show that TS-RAG can still provide meaningful improvements without in-domain data. Furthermore, the results in Table 2 of our paper confirm that TS-RAG remains effective when using a pre-prepared multi-domain knowledge base (without access to the in-domain data).
> > >
> > > ¹ *We evaluated two additional retrieval settings—distribution shift (e.g., using ETTh1 as the knowledge base for ETTh2) and cross-domain (e.g., using weather data as the knowledge base for ETT). As shown in https://shorturl.at/JxB7x **Table 4**, TS-RAG consistently improves performance across all retrieval settings.*
> > >
> > > (2). **Efficiency:** We focus on zero-shot forecasting. Unlike fine-tuning or training new TSFMs which need to tune the model parameters for the target domain, TS-RAG **does not** require model parameter tuning when deployed to a new domain, making it significantly more efficient in terms of time and computational cost.
> > >
> > > (3). **Flexibility:** TS-RAG allows rapid adaptation to changing distributions by simply updating the knowledge base offline, which supports practical and flexible use in real-world scenarios.
> > >
> > > **2. About the pretrained encoder**
> > >
> > > Thank you for the follow-up question. To support this claim, we conducted additional experiments using pretrained encoders from two other TSFMs, i.e., TTM and MOMENT, as the retriever encoder for comparison.
> > >
> > > TTM uses an MLP-Mixer-like architecture, MOMENT is based on a Transformer encoder. The original retriever encoder in our paper is based on Chronos, which is built on a T5 architecture.
> > >
> > > We compared the three encoders under the same setup. As shown in https://shorturl.at/JxB7x **Table 2**, the performance based on all three encoders is comparable, and none of them consistently outperforms the others. This supports our earlier response (Q4), i.e., the choice of encoder architecture (among existing TSFMs) has minor impact on performance.
> > >
> > > **3. Effect of subset size**
> > >
> > > Thank you for raising this question. We would like to clarify this from two perspectives:
> > >
> > > **(1). Subset size of the training set:**
> > >
> > > We conducted experiments with different subset sizes of the pretraining data. As stated in the paper, we constructed a pretraining corpus of 26 million input-output pairs, randomly sampled from the Chronos pretraining dataset. To investigate the effect of data scale, we trained our TS-RAG model on varying proportions of this data—from 0.1% to 50%.
> > >
> > > As shown in https://shorturl.at/JxB7x **Table 3**, the performance of TS-RAG improves as the size of the pre-training dataset grows, but the gains diminish with scale. Specifically, the average MSE (aggregated across 7 datasets used in the paper) improves quickly from 0.1% to 10% of the data, while further improvements from 10% to 50% become minimal.
> > >
> > > **(2). Domain relevance matters more than size of knowledge base**
> > >
> > > **Table 2** in our paper compares retrieval from a large pretraining database (~2.8 million pairs) with much smaller in-domain databases (e.g., ~8 thousand pairs for each sample in ETTh1). Despite its much smaller size, the in-domain database performs better, highlighting the importance of domain relevance. However, a multi-domain knowledge base remains effective when in-domain data is unavailable.
> > >
> > > **4. Zero-shot ability of MOMENT model**
> > >
> > > Thank you for the question. The original MOMENT model does not support zero-shot long-term forecasting due to the lack of a pretrained prediction head. We address this by pretraining a prediction head on the same pretraining data used for TS-RAG. As shown in Table 1 of our paper, MOMENT achieves comparable zero-shot performance to other TSFMs, making the use of MOMENT in TS-RAG both fair and reasonable.
> > >
> > > **5. Inference latency**
> > >
> > > Thank you for the comment. Although retrieval introduces most of the inference latency, we believe the trade-off is rational, as the retrieval-augmented mechanism provides significant improvement in zero-shot performance and improves interpretability.
> > >
> > > More importantly, the overall latency remains at the millisecond level per query, which is acceptable even for real-time applications.
> > >
> > > Finally, the retrieval-augmented forecasting remains relatively underexplored. Our work is a proof-of-concept, focusing on demonstrating the effectiveness rather than optimizing the efficiency. Our future work will explore optimizations such as GPU acceleration, hashing-based indexing to further reduce retrieval latency.

---

### Decision · Program_Chairs · 2025-05-01

**Decision:**

Reject

**Comment:**

The paper explores a promising and interesting direction by applying retrieval-augmented generation (RAG) to time series foundation models for zero-shot forecasting. However, there is concern that the current version of the manuscript does not sufficiently establish the value of this approach. Multiple reviewers noted concerns regarding limited novelty, modest gains over strong baselines such as Chronos-bolt, and substantial added inference cost from retrieval. There are also important missing references and comparisons. Furthermore, the AC is concerned that the paper lacks a critical comparison that is critical for clarifying the full benefit of retrieval. Intuitively, similar improvements could be achieved by simply increasing the historical input length without retrieval. For instance, if we have to choose between retrieving 4 similar sequences vs making the history longer by 4-5 times, it is not clear which method is better and what the incremental value of retrieval is. This omission weakens the empirical claims, as it remains unclear if RAG provides a unique advantage over longer-context baselines. Given these limitations, I recommend rejection in its current form.